# Comparison of Mitochondrial Genome Sequences between Two *Palaemon* Species of the Family Palaemonidae (Decapoda: Caridea): Gene Rearrangement and Phylogenetic Implications

**DOI:** 10.3390/genes14071499

**Published:** 2023-07-22

**Authors:** Yuman Sun, Jian Chen, Yingying Ye, Kaida Xu, Jiji Li

**Affiliations:** 1National Engineering Research Center for Marine Aquaculture, Zhejiang Ocean University, Zhoushan 316022, China; sunyuman@zjou.edu.cn (Y.S.); chenjian@zjou.edu.cn (J.C.); yeyy@zjou.edu.cn (Y.Y.); 2Key Laboratory of Sustainable Utilization of Technology Research for Fisheries Resources of Zhejiang Province, Scientific Observing and Experimental Station of Fishery Resources for Key Fishing Grounds, Ministry of Agriculture and Rural Affairs of China, Zhejiang Marine Fisheries Research Institute, Zhoushan 316021, China

**Keywords:** *Palaemon macrodactylus*, *Palaemon tenuidactylus*, Palaemonidae, mitochondrial genome, gene rearrangement, phylogenetic relationships

## Abstract

To further understand the origin and evolution of Palaemonidae (Decapoda: Caridea), we determined the mitochondrial genome sequence of *Palaemon macrodactylus* and *Palaemon tenuidactylus*. The entire mitochondrial genome sequences of these two *Palaemon* species encompassed 37 typical genes, including 13 protein-coding genes (PCGs), 2 ribosomal RNA genes (rRNAs), and 22 transfer RNA genes (tRNAs), and a control region (CR). The lengths of their mitochondrial genomes were 15,744 bp (*P. macrodactylus*) and 15,735 bp (*P. tenuidactylus*), respectively. We analyzed their genomic features and structural functions. In comparison with the ancestral Decapoda, these two newly sequenced *Palaemon* species exhibited a translocation event, where the gene order was *trnK*-*trnD* instead of *trnD*-*trnK*. Based on phylogenetic analysis constructed from 13 PCGs, the 12 families from Caridea can be divided into four major clades. Furthermore, it was revealed that Alpheidae and Palaemonidae formed sister groups, supporting the monophyly of various families within Caridea. These findings highlight the significant gene rearrangements within Palaemonidae and provide valuable evidence for the phylogenetic relationships within Caridea.

## 1. Introduction

The family Palaemonidae Rafinesque, 1815, belonging to the order Decapoda, infraorder Caridea, and superfamily Palaemonoidea, encompasses numerous economically valuable species and represents one of the largest taxonomic units at the family level within the true shrimp classification [1]. There are approximately 980 species in the world belonging to the family Palaemonidae, with the extant species inhabiting marine, estuarine, and freshwater environments [2]. These diverse habitats have contributed to high physiological, biochemical, morphological, and ecological diversity observed throughout the evolutionary history of the Palaemonidae family. Symbiosis is a widespread and crucial ecological process in nature, contributing significantly to biodiversity [3]. By enabling organisms to receive or exchange mutualistic services and access previously unreachable resources, symbiosis offers opportunities for expanding ecological niches and diversification [4,5]. The family Palaemonidae is one of the biological groups actively engaged in symbiosis, particularly in coral reef ecosystems, displaying a wide array of species, ecological roles, and morphological variations [6]. Some researchers argue that an evolutionary framework would facilitate a comprehensive understanding of the complexity of symbiosis in the Palaemonidae family [3]. However, the scope covered by the Palaemonidae family has been a subject of debate since its establishment. De Grave et al. [2] provided an overview of the most recent classification of the superfamily, and subsequent revisions have relied heavily on molecular techniques surpassing traditional morphological examinations. Multiple independent molecular studies have demonstrated the inclusion of Gnathophyllidae, Hymenoceridae, and Kakaducarididae nested within the Palaemonidae family [7,8,9,10]. These findings have prompted morphological reappraisals aimed at distinguishing family roles. For instance, the studies by Short et al. [11] and De Grave et al. [12] revealed shared morphological characteristics among these three families within the Palaemonidae family, thus considering them synonymous. The taxonomic status of the Palaemonidae family has also been a subject of interest in previous molecular systematic studies of the suborder Caridea. Most studies have proposed a close relationship between the Palaemonidae family and Alpheidae [13,14,15,16,17,18,19], but Li et al.’s research yielded different results [20]. Using five nuclear genes, they constructed both Bayesian inference (BI) and maximum likelihood (ML) phylogenetic trees, which indicated a closer affinity between Alpheidae and Hippolytidae, while Palaemonidae formed a separate branch. Moreover, the internal phylogenetic positions of various families within Caridea have also shown discrepancies in previous molecular systematic studies [13,14,17,18,19].

The rapid advancement of modern molecular biology greatly propelled the research in molecular systematics. By studying molecular sequences to investigate the phylogenetic relationships between species, it was possible to effectively supplement the limitations of traditional taxonomy and address many contentious issues in the fields of classification and systematic evolution [17,21,22]. The mitochondrial genome, characterized by its simple structure, rich genetic information, ease of isolation, and maternal inheritance [23], has been widely utilized in research areas such as population genetic structure, species identification, and systematic evolution [18,24,25]. The expanded availability of complete mitogenomes has the potential to aid in unraveling the phylogeny of Palaemonidae. This can be accomplished by offering multiple loci with varying rates of evolution, thus enhancing our understanding of their evolutionary relationships. In terms of mitochondrial gene arrangement, gene rearrangements are commonly observed in the mitochondrial genomes of crustaceans [20]. Previous studies have also identified non-conservative gene arrangement patterns within the Palaemonidae family, highlighting the necessity of exploring the mitochondrial genome characteristics of this family [1,13,14]. These findings not only contribute to the field of systematic evolution, but also aid in the understanding of the molecular aspects of Palaemonidae. Despite the ecological and economic importance of Palaemonidae species, the available mitogenome data for Palaemonidae is currently quite limited. In GenBank, there are only 18 complete mitogenomes available (until July 5th, 2023, excluding unverified records) (https://www.ncbi.nlm.nih.gov/nuccore).

The *Palaemon macrodactylus* Rathbun, 1902 and *Palaemon tenuidactylus* Liu, Liang & Yan, 1990 belong to the family Palaemonidae. Both species have significant economic value, especially *P*. *macrodactylus*, which had been discussed by American scholar regarding its potential introduction methods [26], other researchers have also documented the expansion of its range [27,28,29]. Studies on *P*. *macrodactylus* mainly focused on its morphology [30], life history [31], ecological behavior [32], the influence of temperature and salinity on larval survival and development [33], and geographical distribution [34,35]. On the other hand, studies on *P*. *tenuidactylus* were relatively scarce, mainly concentrated on its morphology [30] and larval development process [36], with no reports on the complete mitochondrial genomes of these two species. Until now, only 10 species of the genus *Palaemon* with a complete mitogenome available in the GenBank database.

In this study, we conducted the complete mitochondrial genomes of *P*. *macrodactylus* and *P*. *tenuidactylus*. Our objectives are as follows: (1) to enhance taxonomic research methods and provide additional references for the molecular classification of Palaemonidae; (2) to analyze the characteristics and functions of the mitochondrial genomes of two *Palaemon* species, and gain insights into gene function through the assessment of AT skew values and relative synonymous codon usage (RSCU) of protein-coding genes (PCGs); (3) to investigate the patterns of mitochondrial gene rearrangements within Palaemonidae; (4) to elucidate the taxonomic position of the Palaemonidae family within Caridea; (5) to explore the phylogenetic relationships within Caridean shrimp.

## 2. Materials and Method

### 2.1. Sampling, Identifcation and DNA Extraction

Samples of *P*. *macrodactylus* and *P*. *tenuidactylus* were collected from the coastal area of Zhoushan (122°50′ N, 30°09′ E), Zhejiang Province, in the East China Sea. Fresh specimens were preserved in 95% ethanol and transported to the laboratory for subsequent morphological identification by experts from the Marine Biology Museum of Zhejiang Ocean University, with reference to the sixth volume of “An Illustrated Guide to Species in China’s Seas” [37]. Total genomic DNA was extracted from muscle tissue using a salt-extraction method and stored at −20 °C for sequencing [38].

### 2.2. Mitogenome Sequencing, Assembly, and Annotation

The complete mitochondrial genome sequences of *P*. *macrodactylus* and *P*. *tenuidactylus* were sequenced on the Illumina HiSeq X Ten platform by Origin gene Bio-pharm Technology Co. Ltd., Shanghai, China. Genomic DNA of the sample was first quality-checked, and after passing the quality control, 1 μg DNA was used to construct the library. The DNA was randomly fragmented into 300–500 bp fragments using a Covaris M220 ultrasonic disruptor, followed by end-repair, A-tailing, adapter ligation, purification, and PCR amplification to complete the library preparation, and each library produced approximately 10 Gb of raw data. The constructed library was sequenced using the sequencing by synthesis (SBS) technique with an Illumina HiSeq X Ten platform. After the trimming and quality control of the raw data using Cutadapt software [39], the preliminary assembly results were obtained using GetOrganelle (https://github.com/Kinggerm/GetOrganelle (accessed on 9 May 2023)) [40]. The best assembly results were obtained through multiple rounds of correction and iteration. The stack cluster was compared with the genomes of other *Palaemon* species in the GenBank and mitogenomic sequences were verified by checking the *cox1* and *16S rRNA* sequences using NCBI BLAST (https://blast.ncbi.nlm.nih.gov/Blast.cgi (accessed on 9 May 2023)) [41]. Similar codons in other invertebrate species were compared to identify aberrant start and stop codons. Structural and functional annotation was performed using the online software MITOS (http://mitos2.bioinf.uni-leipzig.de/index.py (accessed on 10 May 2023)) [42] and manual corrections were made to obtain the final complete mitogenome. Finally, the sequenced mitogenomes were uploaded to the GenBank database at the National Center for Biotechnology Information (NCBI). The GenBank accession numbers for *P*. *macrodactylus* and *P*. *tenuidactylus* are OQ512152 and OP650931, respectively.

### 2.3. Sequence Analysis

The complete mitogenome was annotated using the Sequin software (version 15.10, http://www.ncbi.nlm.nih.gov/Sequin/ (accessed on 14 May 2023)). NCBI-BLAST was employed to determine the boundaries of protein-coding and ribosomal RNA genes. The correctness of transfer RNA genes and their secondary structures were verified using MITOS WebServer [42]. The base composition was analyzed using DAMBE 7 [43], while the nucleotide composition and relative synonymous codon usage (RSCU) of each protein-coding gene were calculated using MEGA-X [44]. To estimate the strand asymmetry, the formulas AT-skew = (A − T)/(A + T); GC-skew = (G − C)/(G + C) [45] were utilized. Additionally, the circular visualization of the mitogenomes of *P. macrodactylus* and *P. tenuidactylus* was performed using the CGView server (https://cgview.ca/ (accessed on 14 May 2023)) [46]. ORFfinder (https://www.ncbi.nlm.nih.gov/orffinder/ (accessed on 2 May 2023)) was used to find the ORFs (open reading frames) and determine the boundaries between genes.

### 2.4. Gene Order Analysis

In the mitochondrial genomes of Malacostraca, gene rearrangement is commonly observed [47]. The arrangement of mitochondrial genes serves as an important tool in systematic biogeography and phylogenetics, providing significant insights into the evolution of metazoans [48,49]. Currently, four models are primarily used to explain mitochondrial genome rearrangement: (1) duplication–random loss model, where genes are duplicated and individual copies are randomly lost or deleted; (2) tRNA gene-initiated replication errors, where replication starts at a tRNA and is retained as an incorrect replication origin, leading to gene rearrangement; (3) recombination, where gene order changes upon reconnection after double-strand breaks in the DNA; (4) replication–nonrandom loss, where gene duplication forms a dimer, and the loss of transcription promoter function in one set of the dimers leads to directional nonrandom loss or deletion of genes [50].

In addition to the two newly sequenced *Palaemon* species mitochondrial genomes sequenced in this study, we obtained an additional 18 complete Palaemonidae mitochondrial genomes from GenBank (Table 1) for comparative analysis. The gene arrangements of these 20 mitochondrial genomes were compared with the ancestral Decapoda [13,14] in order to investigate the gene rearrangement patterns within the Palaemonidae family. To ensure that observed differences in gene arrangement were not attributed to misannotations, any Palaemonidae mitogenomes that deviated from the ancestral pattern underwent reannotation using MITOS.

### 2.5. Phylogenetic Analysis

To explore the phylogenetic relationships within the Palaemonidae family, we downloaded sequences of 89 species from 12 Caridea families from GenBank (Table 1). We used the mitogenomes of *Solenocera crassicornis* (MF379621) and *Metapenaeopsis dalei* (NC_029457) from Dendrobranchiata as outgroups, and analyzed the phylogenetic relationships based on the 13 PCGs of these 93 species. We used DAMBE 7 software [43] to identify the sequence of the 13 PCGs from each downloaded sample. The nucleotide sequences for all 13 PCGs were individually aligned using the default settings of ClustalW [44] in MEGA X, and then concatenated by PhyloSuite [51]. Afterward, Gblocks v.0.91b [52] was employed with default parameter settings to remove divergent and ambiguously aligned blocks, selecting conserved regions. The substitution saturation was calculated using the GTR substitution model via DAMBE 7, and the third position of the codons was excluded from subsequent analyses due to saturation. We tested the selected DNA sequences for nucleotide models using jModelTest2.1.7 (https://code.google.com/p/jmodeltest2/ (accessed on 15 May 2023)) [53].

We employed two methods to analyze the phylogenetic relationships: the maximum likelihood (ML) method using IQ-tree 2.1.3 [54], and the Bayesian inference (BI) method using MrBayes 3.2.7a [55]. Two partitions (first and second codon positions of 13PCGs) were set in the combined data set for partitioned Bayesian analyses using MrBayes v.3.2, we used PAUP 4 [56] for format conversion, and then used a combination of PAUP 4, ModelTest 3.7 [57], and MRModelTest 2.3 [58] software in MrMTgui to determine the best alternative model (GTR + I + G) according to the Akaike information criterion (AIC). The BI tree analysis was performed using four Markov Chain Monte Carlo (MCMC) chains simultaneously running for 2 million generations, with a sampling frequency of every 1000 generations. In the first burn-in, 25% of trees were discarded, and convergence for independent operation was evaluated using the mean standard deviation of the splitting frequency (<0.01). All parameters for effective sample size (ESS) were checked using Tracer v.1.7 [59]. For ML tree building with IQ-TREE, the same dataset was used. We used ModelFinder [60] to select the best alternative model (TIM2+F+R10) for the ML tree based on the Bayesian Information Criterion (BIC). The consensus tree was reconstructed, and 1000 ultrafast likelihood bootstrap replicates were utilized. Finally, we edited the phylogenetic tree using FigTree v1.4.3 [61].

## 3. Results and Discussion

### 3.1. Genome Structure, Composition, and Skewness

The complete mitochondrial genome sequences of *P. macrodactylus* and *P. tenuidactylus* were 15,744 bp and 15,735 bp, respectively (GenBank accession numbers OQ512152 and OP650931) (Figure 1). The mitogenomes of *both P. macrodactylus* and *P. tenuidactylus* are closed circular double-stranded DNA molecules that contain 37 typical genes, including 13 PCGs, 22 transfer RNA genes (tRNAs), 2 ribosomal RNA genes (rRNAs), and a control region (CR). In both mitogenomes, 23 genes were located on the heavy chain, which contained 9 PCGs (*cox1*, *cox2*, *atp8*, *atp6*, *cox3*, *nad3*, *nad6*, *cytb* and *nad2*) and 14 tRNA genes (*trnL2*, *trnK*, *trnD*, *trnG*, *trnA*, *trnR*, *trnN*, *trnS1*, *trnE*, *trnT*, *trnS2*, *trnI*, *trnM* and *trnW*), while the other 14 genes were located on the light chain (Table 2). The CR was located between *12S rRNA* and *trnI* in both of them, with a length of 180 bp for *P*. *macrodactylus* and 200 bp for *P*. *tenuidactylus* (Table 2).

Both *P. macrodactylus* and *P. tenuidactylus* have 11 gene overlaps in their complete mitogenomes, as well as 16 and 17 gene gaps, respectively (Figure 1, Table 2). The largest intergenic spaces of the two newly sequenced mitogenomes are 439 bp and 506 bp, respectively, located between the CR and *trnI* genes. Additionally, there are two relatively large intergenic regions, measuring 331 bp and 242 bp, respectively, located between the *12S rRNA* and CR. Additionally, the maximum gene overlap in both mitogenomes is of 40 bp, between *trnL1* and *16S rRNA* (Table 2).

The A+T content of the whole mitogenome if 68.06% for *P*. *macrodactylus* (35.77% A, 32.29% T, 11.78% G and 20.17% C), and 73.53% for *P. tenuidactylus* (36.85% A, 36.68% T, 9.94% G and 16.53% C) (Figure 2A). Both two newly sequenced mitogenomes exhibit a high AT bias, with AT-skew values of 0.051 (*P. macrodactylus*) and 0.002 (*P. tenuidactylus*), and GC-skew values of −0.262 (*P. macrodactylus*) and −0.248 (*P. tenuidactylus*), respectively (Figure 2B).

### 3.2. Protein-Coding Genes and Codon Usage

The total lengths of the PCGs in the *P. macrodactylus* and *P. tenuidactylus* mitogenomes were 11,101 bp and 11,127 bp, respectively. The A+T contents were 65.96% and 71.89% for *P. macrodactylus* and *P. tenuidactylus*, respectively, with AT-skews of −0.156 and −0.173 (Figure 2B), indicating a clear bias towards T. The lengths of individual PCGs in both *Palaemon* species were consistent, except for the *nad4* gene, as was their overlap. The longest PCG in both species was the *nad5* gene, at 1713 bp, while the shortest was the *atp8* gene, at 159 bp (Table 2). In both *Palaemon* species, the *atp8* and *atp6* genes overlapped by seven nucleotides, *atp6* and *cox3* overlapped by one nucleotide, and *nad6* and *cytb* overlapped by one nucleotide. The *nad4* and *nad4l* genes of *P. tenuidactylus* overlapped by seven nucleotides (Table 2). Upon comparing the initiation and termination codons of all PCGs in the two *Palaemon* species, we identified four initiation codons and five termination codons (Table 2). Most PCGs in both mitogenomes started with ATG, ATT, and ATA, except for the *atp8* gene in *P. tenuidactylus*, which started with ATC. The *cox1*, *cox2*, *atp6*, *cox3*, *nad4l*, and *cytb* genes in both mitogenomes, as well as the *atp8* gene in *P. macrodactylus* and the *nad4* gene in *P. tenuidactylus*, all initiated with ATG. The *nad3*, *nad5*, *nad2*, and *nad6* genes in both mitogenomes started with ATT, while the *nad1* gene in both mitogenomes and the *nad4* gene in *P. macrodactylus* started with ATA. The majority of the PGCs of the two mitogenomes were terminated with TAA and TAG, except for the *cox1*, *cox2*, *cytb* genes in *P. macrodactylus*, which stopped with CTA, ACT and ATT, respectively. Meanwhile, the *cox1*, *cox2*, *cytb* genes of *P. tenuidactylus* stopped with single T. Incomplete termination codons are a remarkably common phenomenon in mitochondrial genes of vertebrates and invertebrates [62].

Using MEGA-X, the amino acid content (Figure 3A) and RSCU (Figure 3B) of the two *Palaemon* mitogenomes were analyzed. The analysis revealed a relatively similar composition of amino acids in the PCGs for both species. Among the amino acids, Leu1, Lys, and Phe were the most frequently observed, while Arg and Cys were the least common ones. Analysis of codon preference showed that *P*. *macrodactylus* had 34 preferred codons (RSCU ≥ 1) out of the 13 PCGs, whereas *P. tenuidactylus* had 33 preferred codons out of the same number of genes. In the mitogenome of *P. macrodactylus*, the most frequently used codons, in descending order, were UUA (Leu), GGA (Gly), GUA (Val), and CGA (Arg). In the genome of *P. tenuidactylus*, the most frequently used codons, in descending order, were UUA (Leu), GUA (Val), ACU (Thr), and UCU (Ser). Both species had the lowest RSCU values for the codon GCG (Ala). In *P. macrodactylus*, except for the codons UGU (Cys), CGU (Arg), and AGU (Ser), all codons with A or U as the third base had RSCU values greater than 1. Similarly, in *P. tenuidactylus*, except for the codons CCA (Pro), GCA (Ala), AGU (Ser), and CGU (Arg), all codons with A or U as the third base had RSCU values greater than 1. Both species exhibited a preference for using the bases A and T in their codon usages.

### 3.3. Transfer and Ribosomal RNAs

In line with other Caridea mitogenomes, the mitogenomes of the two *Palaemon* species contained 22 tRNA genes. The total lengths of tRNAs in the mitogenomes of *P. macrodactylus* and *P. tenuidactylus* were 1442 bp and 1411 bp, respectively. The length of tRNAs in these species ranged from 61 to 68 bp (Table 2). Apart from the *trnD* and *trnR* genes, the lengths of the other tRNA genes were consistent between the two species. All of the tRNA genes exhibited a high AT content, with *P. macrodactylus* having an AT content of 66.99% and *P. tenuidactylus* having an AT content of 73.78% (Figure 2A). The tRNA genes of *P. macrodactylus* and *P. tenuidactylus* displayed positive AT-skew (0.022 and 0.025, respectively) and GC-skew (0.138 and 0.145, respectively) (Figure 2B). The secondary cloverleaf structure of the 22 tRNAs from these species was examined. In both species, the *trnS1* gene was unable to form a secondary structure due to the lack of dihydrouracil (DHU) arms, which is a common phenomenon in metazoans [63]. Similarly, *trnA*, *trnF*, *trnM*, and *trnT* were also unable to form a secondary structure due to the lack of a TΨC loop. The phenomenon of tRNA gene loss of the TψC loop has also been observed in the genomes of some previous metazoans [13,19,64]. However, the remaining tRNAs of two species were capable of folding into a typical cloverleaf structure (Figure 4). The secondary structure of tRNA allows for base mismatches, and all 22 tRNAs in the two *Palaemon* species exhibited four types of base mismatches. Among them, the G-U mismatch was the most common, with *P*. *macrodactylus* having 36 G-U mismatches and *P*. *tenuidactylus* having 35 G-U mismatches. Both *Palaemon* species had four A-A mismatches, three U-U mismatches, and five A-C mismatches. Except for *trnF*, *trnH*, *trnL1*, *trnT*, *trnV*, *trnW*, and *trnY*, 15 tRNAs showed consistent patterns of base mismatches in both species.

The total lengths of the *16S rRNA* and *12S rRNA* genes were found to be comparable between two species, as *P. macrodactylus* and *P. tenuidactylus* had total lengths of 1337 bp and 1308 bp for *16S rRNA*, and 803 bp and 801 bp for *12S rRNA*, respectively (Table 2). The *16S rRNA* and *12S rRNA* genes of two species were situated between *trnL1* and *trnI*, separated by *trnV*. These genes exhibited high AT contents, with *P. macrodactylus* having an AT content of 73.04% and *P. tenuidactylus* having an AT content of 75.74% (Figure 2A). Additionally, both the rRNA genes of *P. macrodactylus* and *P. tenuidactylus* demonstrated negative AT-skew (−0.059 and −0.015, respectively) and positive GC-skew (0.285 and 0.398, respectively) (Figure 2B).

### 3.4. Gene Rearrangement of the Family Palaemonidae

This study compares the gene arrangement patterns of 20 species in the family Palaemonidae with the ancestral gene arrangement pattern of Decapoda mitochondrial genomes. Among the five genera and twenty species in Palaemonidae (Table 1), five gene arrangement patterns were identified, with four patterns showing variation compared to the ancestral Decapoda (Figure 5). In the two newly sequenced species (*P. macrodactylus* and *P. tenuidactylus*) of this study, the gene arrangement order was consistent with the majority of *Palaemon* species, but transpositions of *trnT* and *trnP* were observed compared to the ancestral gene sequence. This transposition of gene blocks is rare in shrimp but common in crabs. Previous studies have suggested that the transposition of *trnT* and *trnP* may be a shared phenomenon among *Palaemon* species [1,13]. However, with the enrichment of the GenBank database, it was discovered that the gene order in *P*. *modestus* is consistent with the ancestral sequence, providing new references for the gene arrangement patterns in *Palaemon* species. The gene arrangement sequences of four species in the genus *Macrobrachium* (*M*. *nipponense*, *M*. *rosenbergii*, *M*. *bullatum*, *M*. *lanchesteri*) were consistent with the ancestral sequence, indicating a conservative pattern in the evolution of *Macrobrachium* species [14]. *A*. *brevicarpalis* showed transposition events in *16S rRNA* and *trnV*. The mitochondrial genome of *H*. *picta* exhibited a novel order where the gene fragment (*nad1*-*trnL1*-*16S rRNA*-*trnV*-*12S rRNA*-*trnI*-*trnQ*) was moved from downstream of *trnS2* to the position downstream of *nad4l* [13]. *A*. *australis* exhibited the rare absence of the *trnL2* gene in its mitogenome, along with transposition of the *trnL1* gene with *16S rRNA* gene, and transposition of the *trnW* gene with the gene block (*trnC*-*trnY*) [1].

Among the 20 species in Palaemonidae, 15 species showed gene rearrangements compared to the ancestral gene sequence, indicating that gene arrangement is not conserved in Palaemonidae species. However, further systematic analysis is hindered by the limited availability of data for certain genera in the GenBank, such as *Ancylocaris*, *Hymenocera*, and *Anchistus*. More mitochondrial sequences of Palaemonidae species are needed in the future to explore the evolutionary relationships within this family.

### 3.5. Phylogenetic Relationships

Two Caridea phylogenetic trees were constructed using the sequences of 13 PCGs from the mitochondrial genomes, employing BI and ML methods (Figure 6). The analysis included 89 species of Caridea shrimp, with a focus on *P*. *macrodactylus* and *P*. *tenuidactylus*, and using *S*. *crassicornis* (MF379621) and *M*. *dalei* (NC_029457) as outgroups for reference (Table 2). The topologies of the phylogenetic trees constructed using the two methods showed slight differences, primarily manifested in the varying relationships between certain families. There were also subtle disparities in the support values of some branch nodes in both trees. The support values obtained by BI were generally higher than those obtained by ML, with most nodes having a support value of 1. The ML tree revealed the internal phylogenetic relationships among Caridea families as ((((Acanthephyridae + Oplophoridae) + Alvinocarididae) + Nematocarcinidae) + Atyidae) + Pandalidae) + Palaemonidae + Alpheidae) + (((Hippolytidae + Rhynchocinetidae) + Lysmatidae) + Thoridae)))))))), whereas the phylogenetic relationships inferred from the BI tree differed from the ML tree only in the case of four families: Hippolytidae, Rhynchocinetidae, Lysmatidae, and Thoridae. In the BI tree, the relationships of these four families were (((Hippolytidae + Rhynchocinetidae) + Thoridae) + Lysmatidae))). In order to better explore the phylogenetic relationships among the families within Caridea, we compiled the research findings of previous studies and represented the simplified phylogenetic relationships of these families in Figure 7, including A [13], B [14], C [15,16], D&E [18], F [19], G [17], H [20] (Figure 7).

The phylogenetic trees constructed using two methods revealed the evolutionary relationships within Caridea, at the species level, the newly sequenced *Palaemon* species showed a close relationship with *P*. *gravieri*, forming a clade of ((*P. tenuidactylus* + *P. gravieri*) + *P. macrodactylus*). Subsequently, the remaining *Palaemon* species clustered together, indicating a strong monophyly within the genus *Palaemon*. At the genus level, the observation revealed that *Typhlatya* and *Caridina* of Atyidae, as well as *Plesionika* of Pandalidae, were polyphyletic. Both *Palaemon* and *Macrobrachium* of Palaemonidae formed monophyletic groups. In the ML tree, their internal relationships were represented as ((*Palaemon* + *Macrobrachium*) + ((*Hymenocera* + *Ancylocaris*) + *Anchistus*)), while in the BI tree, they were depicted as ((*Palaemon* + *Macrobrachium*) + ((*Anchistus* + *Ancylocaris*) + *Hymenocera*)). Previous research conducted by Chow et al. employed four PCGs (*H3*, *Enol*, *GAPDH*, *Nak*) and three rRNA genes (*16S*, *12S*, *18S*) to construct ML and BI phylogenetic trees [6]. The topologies of the two trees were different, as their analysis included more Palaemonidae species. Their results suggested that *Macrobrachium* displayed a polyphyletic pattern, indicating the need for further verification regarding its monophyly. Additionally, their trees only included the four genera relevant to this study, and the inferred relationships among these genera were consistent with the present research. The discrepancies in topology between the two trees may arise from the different computational methods employed [65].

At the family level, species from each family formed a distinct clade, demonstrating the good monophyly of these families. These families then formed four major branches. The first major branch consisted of five families, with Acanthephyridae and Oplophoridae forming a sister group, subsequently clustering with Alvinocarididae, then with Nematocarcinidae, and finally merging with Atyidae to form a larger branch. This result is consistent with previous studies that used 13 PCGs to construct phylogenetic trees [18,19] (Figure 7D,F). It is worth noting that the topologies of the BI and ML trees also differed slightly from those in Chak et al.’s study [18] (Figure 7D,E). Their BI tree supported the relationships among these five families, whereas the ML tree only supported the relationship of (((Acanthephyridae + Oplophoridae) + Alvinocarididae) + Nematocarcinidae), with Atyidae in a separate lineage. Furthermore, Wang et al.’s results also supported the relationships of these four families [17] (Figure 7G). However, there is still some controversy regarding the evolutionary position of Atyidae. Specifically, our results conflict with those of Li et al. [20], who suggested that Atyidae represent basal lineages within Caridea based on five nuclear genes (*Enolase*, *H3*, *NaK*, *PEPCK*, *18S rRNA*) [15]. Similarly, Bracken et al. inferred that Atyidae represent basal lineages within Caridea based on both mitochondrial and nuclear genes [66]. These differences may be due to the heterogeneity of the samples used. The second major branch represented the family Pandalidae, which was also supported by previous studies as monophyletic [13,14,17,18,19] (Figure 7A,B,D,G). The third major branch consisted of the families Palaemonidae and Alpheidae, which formed a sister group. This relationship was also validated in previous studies [13,14,15,16,17,18,19] (Figure 7A–G). However, Li et al.’s study [20] suggested a closer affinity between the families Alpheidae and Hippolytidae, indicating differences possibly due to the heterogeneity of the samples used (Figure 7H). Meanwhile, Li et al. [20] suggested that the family Palaemonidae was not a monophyletic group. Their study indicated that members of the families Hymenoceridae and Gnathophyllidae were clustered within the Palaemonidae. However, according to the latest records from WoRMS, the families Hymenoceridae and Gnathophyllidae have been updated to Palaemonidae (https://www.marinespecies.org/aphia.php?p=taxdetails&id=106788 (accessed on 22 May 2023)). The fourth major branch in both phylogenetic trees comprised the families Hippolytidae, Rhynchocinetidae, Lysmatidae, and Thoridae. Both results supported the closest relationship between Hippolytidae and Rhynchocinetidae. In the ML tree, these two families were closest to Lysmatidae, while in the BI tree, they were closest to Thoridae. This difference may be attributed to the distinct computational methods used [65]. However, our research results regarding the fourth major branch do not align with Cronin et al.’s study [19], where the family Rhynchocinetidae formed a separate branch, while the families Hippolytidae, Lysmatidae, and Thoridae were most closely related (Figure 7F).

Considering the aforementioned research findings, the phylogenetic relationships within Caridea still pose certain questions due to the uneven representation of Caridea species data in GenBank. Some families, such as Rhynchocinetidae, Hippolytidae, Thoridae, Nematocarcinidae, and Oplophoridae, have limited mitochondrial genomic data. Future studies on Caridea phylogeny should incorporate more species from these relevant families to validate and support previous research findings.

## 4. Conclusions

We sequenced the complete mitochondrial genomes of two *Palaemon* species and analyzed the fundamental characteristics of their gene sequences. We found that the genome size and nucleotide composition were similar to those in previous research findings. Among the 22 tRNA genes in these two species, the *trnS1* gene was unable to form a secondary structure due to the absence of the DHU arm. Similarly, the *trnA*, *trnF*, *trnM*, and *trnT* genes were also unable to form secondary structures due to the absence of the TψC loop. However, the remaining tRNA genes in both species were able to fold into the typical cloverleaf structure. The gene arrangement in both *Palaemon* species underwent the same rearrangement pattern compared to the ancestral gene order of Decapoda, with a reversal occurring in the position of *trnK*-*trnD*. Additionally, a comparison of the gene arrangement patterns within Palaemonidae revealed a significant occurrence of extensive gene rearrangements. Phylogenetic analysis demonstrated that species from the 12 families formed separate clades, exhibiting a good level of monophyly. Our research results supported the division of these families into four major clades. Phylogenetic analysis also indicated that Acanthephyridae and Oplophoridae were sister groups, clustering with Alvinocarididae, while Alpheidae and Palaemonidae were sister groups. This study provides extensive information regarding the mitogenomes of *Palaemon*, laying a solid foundation for future research into genetic variation, systematic evolution, and breeding of *Palaemon* using mitogenomes.

## Figures and Tables

**Figure 1 genes-14-01499-f001:**
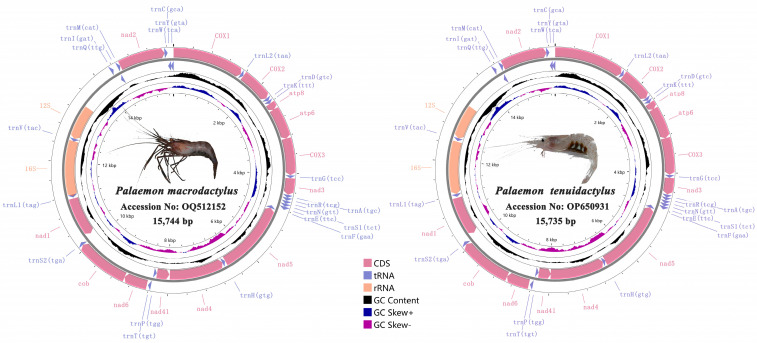
Complete mitogenome map of *P*. *macrodactylus* and *P*. *tenuidactylus*.

**Figure 2 genes-14-01499-f002:**
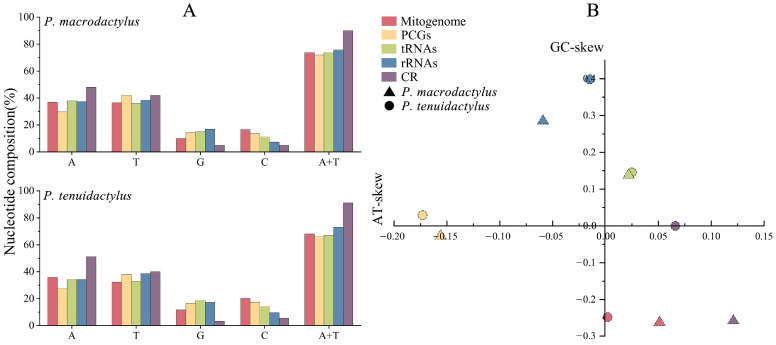
Nucleotide composition of *P*. *macrodactylus* and *P*. *tenuidactylus* mitochondrial genome (**A**). Nucleotide skews of the different gene types within the mitochondrial genomes of *P*. *macrodactylus* and *P*. *tenuidactylus* (**B**).

**Figure 3 genes-14-01499-f003:**
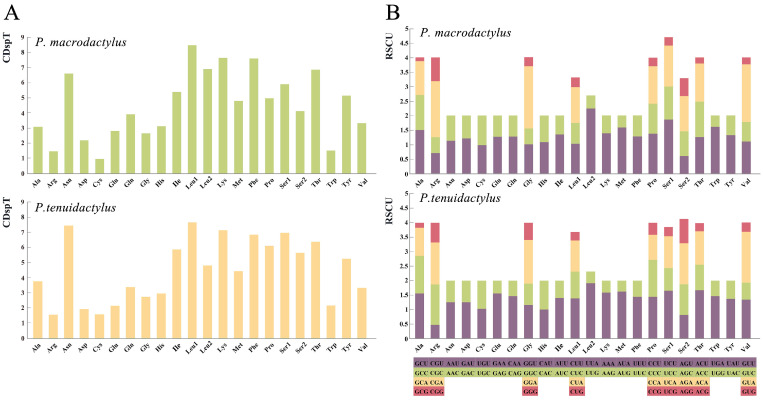
The frequency of mitochondrial PCG amino acids (**A**) and relative synonymous codon usage (RSCU) (**B**) of the two newly sequenced *Palaemon* mitogenomes.

**Figure 4 genes-14-01499-f004:**
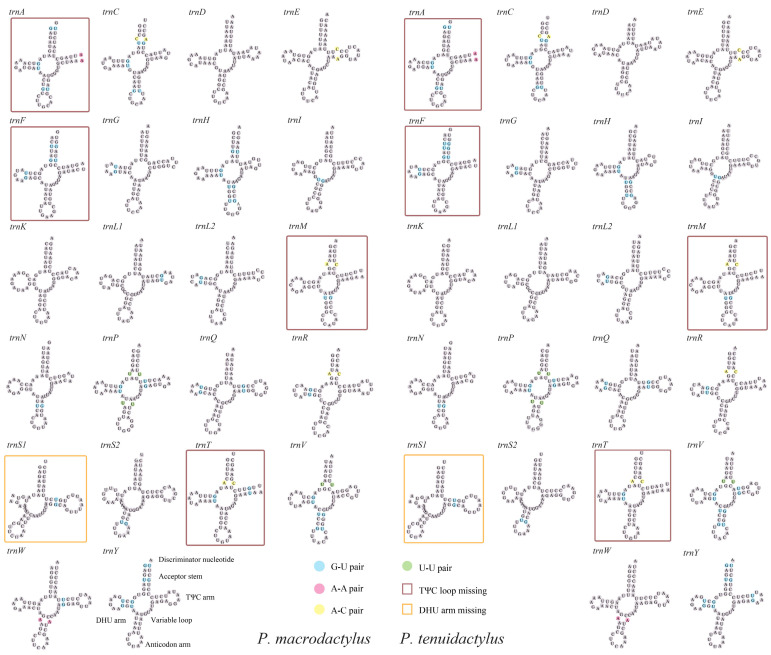
Secondary structures of tRNAs of the two newly sequenced *Palaemon* mitogenomes.

**Figure 5 genes-14-01499-f005:**
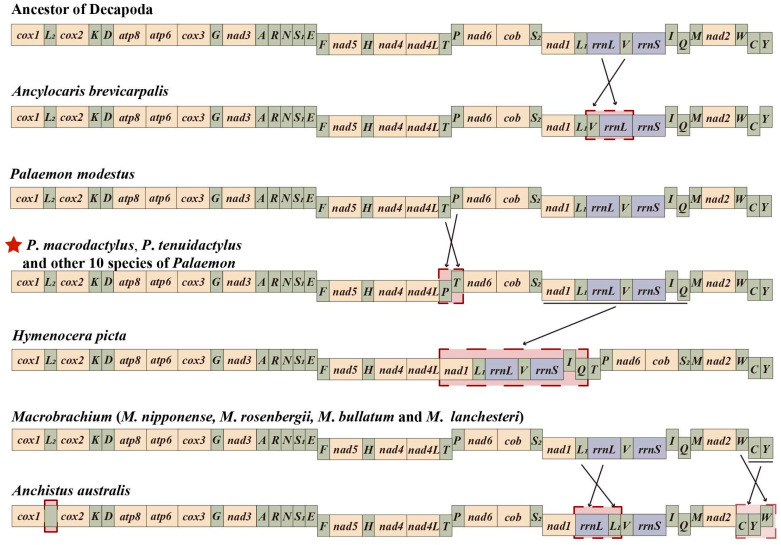
Linear representation of the mitochondrial gene arrangement of the ancestral mitogenome of pancrustaceans and Palaemonid species. In this study, the two newly sequenced species with gene rearrangements are marked with red star, and the rearranged gene blocks are signed by red gridlines and compared with the gene arrangement of ancestral Caridea.

**Figure 6 genes-14-01499-f006:**
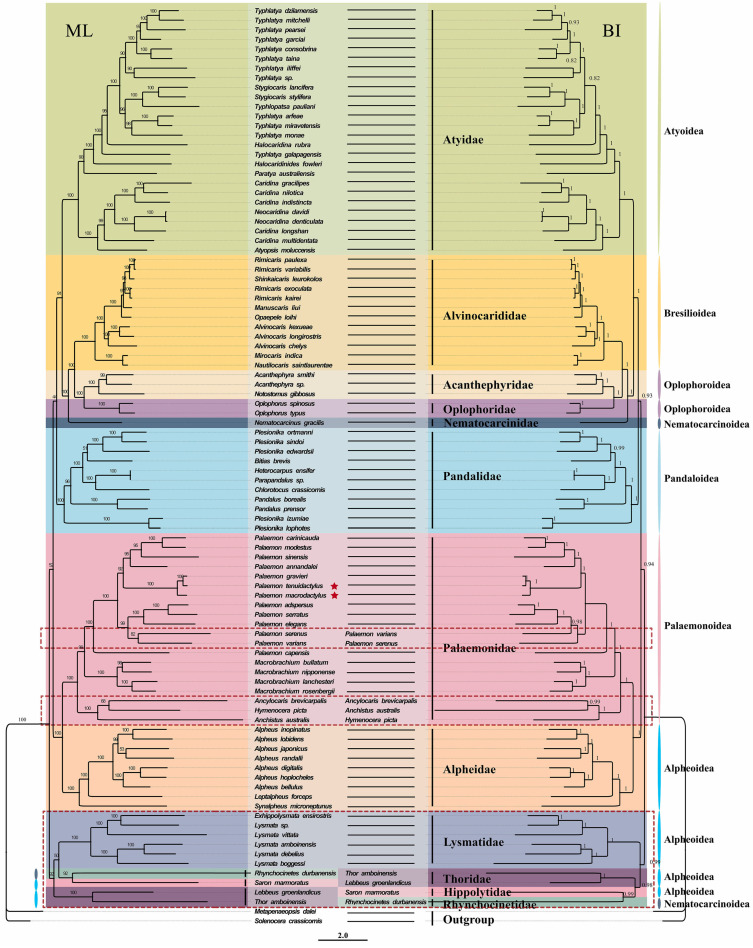
The phylogenetic tree based on 13 PCGs was inferred using Bayesian inference (BI) and maximum likelihood (ML) methods. The species with the same branches in the ML tree and the BI tree are replaced with black horizontal lines, and only species with different branches are displayed. The number at each branch is the bootstrap probability, and the two newly sequenced species are marked with red stars. Nodes in the ML tree with bootstrap support lower than 70 have been collapsed.

**Figure 7 genes-14-01499-f007:**
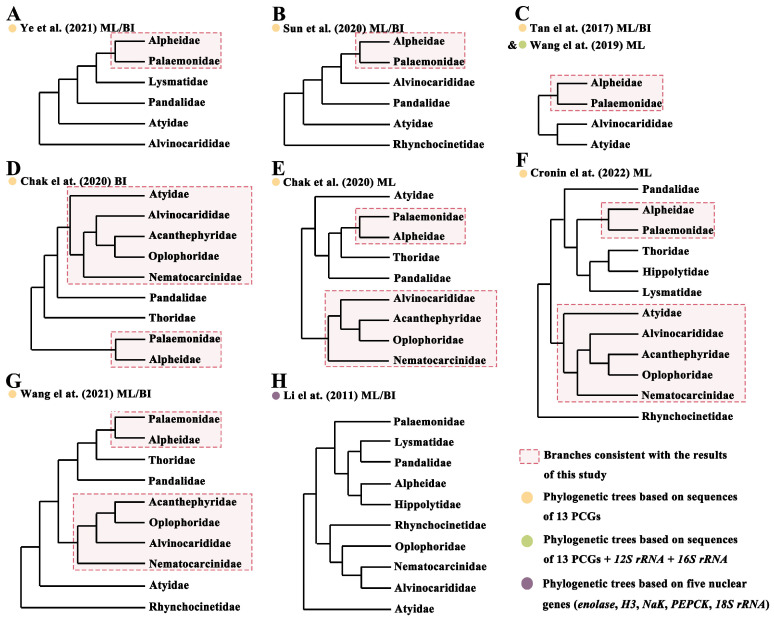
Previous results of phylogenetic studies based on molecular data. Figures (**A**–**H**) represent the results of Caridea phylogenetic trees constructed by different scholars based on different molecular sequences or different species, including (**A**) [13], (**B**) [14], (**C**) [15,16], (**D**,**E**) [18], (**F**) [19], (**G**) [17], (**H**) [20].

**Table 1 genes-14-01499-t001:** List of species analyzed in this study and their GenBank accession numbers, and two newly sequenced *Palaemon* species were marked with *.

Subfamily	Family	Species	Size (bp)	GenBank
Alpheoidea	Alpheidae	*Alpheus digitalis*	15,700	NC_014883
		*Alpheus hoplocheles*	15,735	NC_038068
		*Alpheus inopinatus*	15,789	NC_041151
		*Alpheus japonicus*	16,619	NC_038116
		*Alpheus bellulus*	15,738	MH796167
		*Alpheus lobidens*	15,735	KP276147
		*Alpheus randalli*	15,676	MH796168
		*Synalpheus microneptunus*	15,603	NC_047307
		*Leptalpheus forceps*	15,463	MN732884
	Lysmatidae	*Lysmata amboinensis*	16,735	NC_050676
		*Lysmata boggessi*	17,345	NC_064049
		*Lysmata* sp.	16,758	MW836830
		*Lysmata debelius*	16,757	NC_060421
		*Lysmata vittata*	22,003	NC_049878
		*Exhippolysmata ensirostris*	16,350	MK681888
	Thoridae	*Thor amboinensis*	15,553	NC_051930
		*Lebbeus groenlandicus*	17,399	NC_045223
	Hippolytidae	*Saron marmoratus*	16,330	NC_050677
Atyoidea	Atyidae	*Stygiocaris lancifera*	15,787	NC_035404
		*Stygiocaris stylifera*	15,812	NC_035411
		*Typhlatya arfeae*	15,887	NC_035410
		*Typhlatya consobrina*	15,758	NC_035407
		*Typhlatya dzilamensis*	15,892	NC_035408
		*Typhlatya galapagensis*	16,430	NC_035402
		*Typhlatya garciai*	15,318	NC_035409
		*Typhlatya iliffei*	15,926	NC_035401
		*Typhlatya miravetensis*	15,865	NC_036335
		*Typhlatya mitchelli*	15,814	NC_035403
		*Typhlatya monae*	16,007	NC_035405
		*Typhlatya pearsei*	15,798	NC_035400
		*Typhlatya taina*	15,790	NC_035399
		*Typhlopatsa pauliani*	15,824	NC_035406
		*Typhlatya* sp.	15,870	KX844713
		*Caridina gracilipes*	15,550	NC_024751
		*Caridina indistincta*	15,461	NC_039593
		*Caridina longshan*	15,556	OP177695
		*Caridina multidentata*	15,825	NC_038067
		*Caridina nilotica*	15,497	NC_030219
		*Neocaridina davidi*	15,564	MN418055
		*Paratya australiensis*	15,990	NC_027603
		*Halocaridina rubra*	16,065	NC_008413
		*Halocaridinides fowleri*	15,977	NC_035412
		*Atyopsis moluccensis*	15,933	NC_070241
		*Neocaridina denticulata*	15,561	NC_023823
Palaemonoidea	Palaemonidae	*Palaemon adspersus*	15,736	NC_050168
		*Palaemon annandalei*	15,718	NC_038117
		*Palaemon capensis*	15,925	NC_039373
		*Palaemon gravieri*	15,740	NC_029240
		*Palaemon serratus*	15,758	NC_050266
		*Palaemon serenus*	15,967	NC_027601
		*Palaemon varians*	14,889	MT340090
		*Palaemon elegans*	15,650	MT340089
		*Palaemon modestus*	15,736	MF687349
		*Palaemon sinensis*	15,736	MN372141
		* *Palaemon tenuidactylus*	15,735	OP650931
		* *Palaemon macrodactylus*	15,744	OQ512152
		*Palaemon carinicauda*	15,730	EF560650
		*Macrobrachium nipponense*	15,806	NC_015073
		*Macrobrachium rosenbergii*	15,772	NC_006880
		*Macrobrachium bullatum*	15,774	KM978918
		*Macrobrachium lanchesteri*	15,694	NC_012217
		*Ancylocaris brevicarpalis*	16,673	NC_061664
		*Anchistus australis*	15,396	NC_046034
		*Hymenocera picta*	15,786	NC_039631
Bresilioidea	Alvinocarididae	*Alvinocaris chelys*	15,910	NC_018778
		*Alvinocaris longirostris*	16,022	NC_042497
		*Alvinocaris kexueae*	15,864	MH714459
		*Rimicaris paulexa*	15,909	NC_051948
		*Mirocaris indica*	15,922	NC_054368
		*Nautilocaris saintlaurentae*	15,928	NC_021971
		*Opaepele loihi*	15,905	NC_020311
		*Rimicaris exoculata*	15,902	NC_027116
		*Shinkaicaris leurokolos*	15,903	NC_037487
		*Manuscaris liui*	15,903	MH714461
		*Rimicaris kairei*	15,900	NC_020310
		*Rimicaris variabilis*	15,909	MN419306
Pandaloidea	Pandalidae	*Bitias brevis*	15,891	NC_040856
		*Chlorotocus crassicornis*	15,935	NC_035828
		*Heterocarpus ensifer*	15,939	NC_040855
		*Pandalus borealis*	15,956	LC341266
		*Pandalus prensor*	17,194	MW091549
		*Parapandalus* sp.	16,037	MH714458
		*Plesionika edwardsii*	15,956	OP087601
		*Plesionika sindoi*	15,908	MH714453
		*Plesionika ortmanni*	15,908	OP650932
		*Plesionika izumiae*	16,074	OP650933
		*Plesionika lophotes*	15,933	OP650934
Oplophoroidea	Acanthephyridae	*Notostomus gibbosus*	17,590	NC_059935
		*Acanthephyra* sp.	16,205	MT879756
		*Acanthephyra smithi*	17,165	MH714455
	Oplophoridae	*Oplophorus spinosus*	17,346	NC_059714
		*Oplophorus typus*	16,883	MH714457
Nematocarcinoidea	Nematocarcinidae	*Nematocarcinus gracilis*	15,919	MH714456
	Rhynchocinetidae	*Rhynchocinetes durbanensis*	17,695	NC_029372

**Table 2 genes-14-01499-t002:** Annotation of the *P*. *macrodactylus* and *P*. *tenuidactylus* complete mitochondrial genomes.

Feature	Strand	*P*. *macrodactylus*					*P*. *tenuidactylus*					Anticodon
		Location		Intergenic Region	Size	Start/Stop Codon	Location		Intergenic Region	Size	Start/Stop Codon	
		form	to				form	to				
*cox1*	+	1	1535	0	1535	ATG/CTA	1	1535	0	1535	ATG/T(AA)	
*trnL2*	+	1536	1600	2	65		1536	1600	2	65		TAA
*cox2*	+	1603	2290	3	688	ATG/ACT	1603	2290	3	688	ATG/T(AA)	
*trnK*	+	2294	2359	2	66		2294	2359	4	66		TTT
*trnD*	+	2362	2426	0	65		2364	2424	2	61		GTC
*atp8*	+	2427	2585	−7	159	ATT/TAA	2427	2585	−7	159	ATC/TAA	
*atp6*	+	2579	3253	−1	675	ATG/TAA	2579	3253	−1	675	ATG/TAA	
*cox3*	+	3253	4041	3	789	ATG/TAA	3253	4041	3	789	ATG/TAA	
*trnG*	+	4045	4108	0	64		4045	4108	0	64		TCC
*nad3*	+	4109	4462	3	354	ATT/TAA	4109	4462	4	354	ATT/TAA	
*trnA*	+	4466	4528	−1	63		4467	4529	−1	63		TGC
*trnR*	+	4528	4595	3	68		4529	4594	3	66		TCG
*trnN*	+	4599	4662	−1	64		4598	4661	−1	64		GTT
*trnS1*	+	4662	4728	0	67		4661	4727	0	67		TCT
*trnE*	+	4729	4796	−2	68		4728	4795	−2	68		TTC
*trnF*	−	4795	4858	0	64		4794	4857	0	64		GAA
*nad5*	−	4859	6571	12	1713	ATT/TAA	4858	6570	12	1713	ATT/TAA	
*trnH*	−	6584	6645	0	62		6583	6644	0	62		GTG
*nad4*	−	6646	7962	11	1317	ATA/TAA	6645	7979	−7	1335	ATG/TAA	
*nad4l*	−	7974	8273	7	300	ATG/TAA	7973	8272	7	300	ATG/TAA	
*trnP*	−	8281	8346	15	66		8280	8345	15	66		TGG
*trnT*	+	8362	8425	26	64		8361	8424	26	64		TGT
*nad6*	+	8452	8955	−1	504	ATT/TAA	8451	8954	−1	504	ATT/TAA	
*cytb*	+	8955	10,089	0	1135	ATG/ATT	8954	10088	0	1135	ATG/T(AA)	
*trnS2*	+	10,090	10,157	29	68		10,089	10,156	26	68		TGA
*nad1*	−	10,187	11,125	27	939	ATA/TAA	10,183	11,121	27	939	ATA/TAA	
*trnL1*	−	11,153	11,218	−40	66		11,149	11,214	−40	66		TAG
*16S*	−	11,179	12,515	−6	1337		11,175	12,482	22	1308		
*trnV*	−	12,510	12,574	−1	65		12,505	12,569	−1	65		TAC
*12S*	−	12,574	13,376	331	803		12,569	13,369	242	801		
CR	+	13,708	13,887	/	180		13,612	13,811	/	200		
*trnI*	+	14,327	14,393	28	67		14,318	14,384	28	67		GAT
*trnQ*	−	14,422	14,489	3	68		14,413	14,480	3	68		TTG
*trnM*	+	14,493	14,558	0	66		14,484	14,549	−15	66		CAT
*nad2*	+	14,559	15,551	−2	993	ATT/TAG	14,535	15,542	−2	1008	ATT/TAG	
*trnW*	+	15,550	15,617	−1	68		15,541	15,608	−1	68		TCA
*trnC*	−	15,617	15,679	0	63		15,608	15,670	0	63		GCA
*trnY*	−	15,680	15,744	0	65		15,671	15,735	0	65		GTA

## Data Availability

All mitogenome sequence data were deposited in Genbank with accession numbers OQ512152 (*P*. *macrodactylus*) and OP650931 (*P*. *tenuidactylus*).

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
