# Peer review of "Comparison of Mitochondrial Genome Sequences between Two Palaemon Species of the Family Palaemonidae (Decapoda: Caridea): Gene Rearrangement and Phylogenetic Implications"

_genes, 2023, doi:10.3390/genes14071499_

Round 1

Reviewer 1 Report

Dear Authors,

I read the manuscript entitled "Comparison of mitochondrial genome sequences between two Palaemon species of the family Palaemonidae (Decapoda: Caridea): Gene Rearrangement and Phylogenetic implications" with great interest, since I cope with molecular phylogenetics myself.

The manuscript is worth considering publication in Genes, but it can be improved both in terms of general interest and in terms of methodology in my opinion. My major concern is about the phylogenetic inference: a partitioned analysis should be carried out on PCGs (and maybe on rRNAs) and precisely detailed in the Materials and method section. Both IQ-tree and MrBayes allow cutting-edge analyses that were not implemented in the present paper. For instance, it is possible to compare the use of nucleotides and amino acids; to set the use of different codon positions; to investigate saturation in data; to contrast different phylogenetic constraints.

Moreover, the manuscript in its current form is a long description of the newly sequenced mtDNAs, while the phylogeny is poorly discussed. I think that the manuscript could be greatly improved by enlarging the phylogenetic section, including a wider discussion and, possibly, a time calibration.

Finally, I attach a file with minor concerns and typos.

Best wishes!

Dear Authors,

I am not an English native speaker, but the English language is well written and clear in my opinion. I found some minor typos, which I report in the aligned file; moreover, an uncommon use of the past tense is present throughout the manuscript.

Best wishes!

Author Response

Response to Editors and Reviews

Manuscript ID: genes-2460626

Article Title: Comparison of mitochondrial genome sequences between two Palaemon species of the family Palaemonidae (Decapoda: Caridea): Gene Rearrangement and Phylogenetic implications

Dear Editors and Reviewers:

Thank you for your comments and suggestions on the paper. We have made efforts to clarify the following points and correct any mistakes. We sincerely appreciate the hard work of the editors and reviewers and hope that these revisions will be approved. On the following pages, please find the original comments from the reviewers (in black font) and our responses (in blue font). The revised manuscript has been completed, and all modifications have been made using the track changes mode. We hope that the revised manuscript meets the criteria for publication.

A point-by-point response to the comments:

Reviewer 1

I read the manuscript entitled "Comparison of mitochondrial genome sequences between two Palaemon species of the family Palaemonidae (Decapoda: Caridea): Gene Rearrangement and Phylogenetic implications" with great interest, since I cope with molecular phylogenetics myself.

The manuscript is worth considering publication in Genes, but it can be improved both in terms of general interest and in terms of methodology in my opinion. My major concern is about the phylogenetic inference: a partitioned analysis should be carried out on PCGs (and maybe on rRNAs) and precisely detailed in the Materials and method section. Both IQ-tree and MrBayes allow cutting-edge analyses that were not implemented in the present paper. For instance, it is possible to compare the use of nucleotides and amino acids; to set the use of different codon positions; to investigate saturation in data; to contrast different phylogenetic constraints.

Moreover, the manuscript in its current form is a long description of the newly sequenced mtDNAs, while the phylogeny is poorly discussed. I think that the manuscript could be greatly improved by enlarging the phylogenetic section, including a wider discussion and, possibly, a time calibration.

Finally, I attach a file with minor concerns and typos.

Major comments:

  1. Both generic names and species names should be italicized throughout the manuscript: see, f.i., l. 20; 24; 131; 152 for generic names; l. 110; 118-119; 149; 187-188; 189-190; 197; 198; 205; 206; 208-210. Gene names should also be italicized: see, f.i., ll. 193-196; Finally, authors should use “cytb” instead of “cob”.

Answer: Thank you for your advice and careful review. We have now italicized all species names, generic names, and gene names, and replaced "cob" with "cytb".

Abstract:

  1. I suggest to turn verbs into the present tense: “Palaemonidae encompass numerous economical valuable species and are widely distributed”…

Answer: Thank you for your advice. We have turned verbs into the present tense: “Palaemonidae encompass numerous economical valuable species and are widely distributed”.

  1. l. 18. I suggest to replace “molecular systematic support” with “support from molecular systematics”.

Answer: We have replaced “molecular systematic support” with “support from molecular systematics”.

Introduction:

  1. l. 45. Again, please turn verbs to the present tense when relevant: “There are approximately 981 species”. See also, f.i., “belonged” (l. 91) and “had” (l. 91).

Answer: We have converted the verbs to the present tense.

  1. l. 144. Maybe a space should be inserted before “[42]”, “[44]” (l. 147), and “[50]” (l. 178).

Answer: Thank you for the reminder. We have inserted spaces in the appropriate positions.

  1. l. 159. There is no need for citing MITOS again.

Answer: We have removed the excess citations.

  1. l. 169. It is not clear whether the “unified sequence” was partitioned for analysis or not. For what follows, it seems that a single model was applied to the entire alignment (l. 175), which should be highly discouraged. IQ-tree implements highly valuable algorithms dealing with heterogeneous datasets that should be properly explored. Alignment can be partitioned using genes and even codon positions, if nucleotides were employed. The same holds for the Bayesian analysis, and authors should clearly state the analysis pipeline, which is quickly touched on (ll. 178-179).

Answer: Thank you for your suggestion. These issues may have arisen due to inaccurate descriptions in our previous version. We have made clearer descriptions in the revised manuscript. Firstly, we started with cox1 and connected the 13 PCGs of each species in the same gene order, creating a smaller dataset. Subsequently, we combined all PCGs from all species into a larger dataset for tree construction. Then, we performed alignment and trimming of the PCG sequences of 93 species using ClustalW in MEGA-X. Next, we calculated nucleotide substitution saturation using DAMBE 7 and assessed the suitability of these sequences for constructing the phylogenetic tree. Furthermore, instead of applying a single model to the entire sequence, we selected the topology with the highest likelihood from 1000 replicates as the final tree. We have also provided a clearer description of the analysis for Bayesian trees.

  1. l. 170. I do not understand this “81”. I would have expected “93”.

Answer: Thank you for your careful review. It was a writing error, and we have now corrected it by changing "81" to "93".

  1. ll. 170-171. This sentence is not clear. What is the “suitability for phylogenetic tree construction”? Authors should properly address saturation in the dataset, for example.

Answer: Thank you for your suggestion. We have rewritten this part in the revised manuscript as follows: " We used DAMBE 7 software to identify the sequence of the 13 PCGs from each sample, and adjust the nucleotide sequence of each PCGs, and the substitution saturation was calculated via the GTR substitution model. Starting with cox1, we concatenated the 13 PCGs of each species in the same gene order to create a small dataset. Then, we concatenated the PCGs of all species into one large dataset for tree construction. We aligned and trimmed the PCG sequences of 93 species using ClustalW in MEGA-X. "

  1. ll. 181-182. Not sure if 2 million generations are sufficient. How was convergence estimated? Was ESS taken into account? It is not clear to me the idea of “aging data” and of “developmental tree”, but the use of an a priori 25% burn-in may not represent the best alternative.

Answer: Thank you for your suggestion. These issues may have arisen due to inaccurate descriptions in our previous version. We have made clearer descriptions in the revised manuscript. The use of 2 million generations for sampling is sufficient. Convergence for independent operations was assessed using the mean standard deviation of the splitting frequency (<0.01), and the first burn-in 25%  of trees were discarded instead of being used.

Results and Discussion

  1. l. 192. I do not understand the use of “All of them” here. Please rephrase.

Answer: We have changed "All of them" to "All of mitogenomes".

  1. ll. 193-196. How was the CR determined? By comparison with available palaemonids? This should be clearly stated.

Answer: We obtained the CR fragments through annotation using MITOS. Meanwhile, we compared them with the mitochondrial genomes of other Palaemonidae species in Genbank and found that they were not annotated for the CR region or presumed control regions. The accession numbers of the reference species are: MG787410, MN372141, NC_061664, NC_050266, NC_015073, NC_006880, EF560650, NC_050266.

(i.e. NC_045090 )

  1. l. 203. CR is not properly a “gene”. Moreover, is it interesting that large noncoding regions are situated upstream and downstream the CR itself. How can authors establish boundaries between the CR and neighboring unassigned regions? How can they be sure that it is not a single CR?

Answer: Thank you for your careful review. In the revised version of Table 2, we have removed the italics formatting for the CR. We predicted the CR region using MITOS and compared it with the mitochondrial genomes of other Palaemonidae species in Genbank. However, due to the complexity of this region and the presence of repetitive sequences, there may be some errors during the sequencing and assembly process, which could lead to potential inaccuracies in subsequent predictions.

  1. l. 209 (and l. 252; 320). Please remove the space after the minus sign.

Answer: We have removed the space after the minus sign.

  1. l. 253. What is a “clear AT bias”? I would say a “clear bias towards T”.

Answer: We have changed " clear AT bias " to " clear bias towards T".

  1. l. 272. MEGA-X was declared at l. 147.

Answer: We have changed " MEGA7" to " MEGA-X".

  1. l. 276. I would add “ones” after “common”.

Answer: We have added “ones” after “common”.

  1. l. 291. “Ribosomal” should not be italicized.

Answer: Thank you for your suggestion, but the italics here are used because the journal's subheadings are consistently formatted in italics.

  1. l. 311. “Except for” does not make sense with “the remaining” in my opinion (if they are “the remaining”, I do not have to eliminate exceptions from them): please rephrase.

Answer: Thank you for your suggestion. We have removed "the remaining" from the sentence.

  1. l. 324. Since the amount of gene rearrangement is somewhat reduced, I suggest to perform a phylogenetic analysis using the gene order on the mitochondrial molecule, which could provide further insights beside phylogenetic inference.

Answer: Thank you for your suggestion, but we believe that arranging the gene sequences of all species in the same order for phylogenetic analysis seems more reliable. Some published articles and our research have employed the same method for phylogenetic analysis, for instance:

  1. Miao J, Feng J, Liu X, et al. Sequence comparison of the mitochondrial genomes of five brackish water species of the family Neritidae: Phylogenetic implications and divergence time estimation. Ecology and Evolution. 2022 Jul;12(6):e8984. DOI: 10.1002/ece3.8984.
  2. Duan X, Dong X, Li J, Lü J, Guo B, Xu K, Ye Y. The Complete Mitochondrial Genome of Pilumnopeus Makianus (Brachyura: Pilumnidae), Novel Gene Rearrangements, and Phylogenetic Relationships of Brachyura. Genes. 2022; 13(11):1943. https://doi.org/10.3390/genes13111943
  3. Sun, Shao'e, Zhong-li Sha and Yan-rong Wang. “Mitochondrial phylogenomics reveal the origin and adaptive evolution of the deep-sea caridean shrimps (Decapoda: Caridea).” Journal of Oceanology and Limnology 39 (2021): 1948 - 1960. DOI:10.1007/s00343-020-0266-4.
  4. Zhang, Y., Wei, L., Liu, B. et al. Two complete mitogenomes of Ocypodoidea (Decapoda: Brachyura), Cleistostoma dilatatum (Camptandriidae) and Euplax sp. (Macrophthalmidae) and its phylogenetic implications. Acta Oceanol. Sin. 42, 81–92 (2023). https://doi.org/10.1007/s13131-022-2054-9.
  5. l. 341. How was the “ancestral Decapoda” gene order determined?

Answer: Previous studies have documented the gene order of the ancestral Decapoda, and we have added the corresponding reference in this section of the article.

  1. l. 417 (and l. 424). I would use a comma instead of the ampersand.

Answer: Thank you for your careful review. We have use a comma instead of the ampersand.

  1. l. 420. Two parentheses are opended, while three are closed.

Answer: Thank you for your careful review. We have translated the original text to: (((Acanthephyridae + Oplophoridae) + Alvinocarididae) + Nematocarcinidae).

  1. l. 426. Only the first letter of “LI” should be capitalized.

Answer: Thank you for your careful review. We have changed " LI" to " Li".

  1. l. 433. Please remove the comma before “they”.

Answer: We have removed the comma before “they”.

Data Availability Statement

  1. The use of “462,” is not clear to me.

Answer: Thank you for your careful review. It was a writing error, we have removed "462".

Figures

  1. Figure 1: Please check the use of the thousand separator.

Answer: Thank you for your careful review. It was a writing error, we have made modifications to Figure 1.

  1. Figure 5: l. 369. Actually, it is not a pentagram (see also Fig. 6 caption), I would say a “star”. Moreover, there should be no stop before “And”.

Answer: Thank you for your careful review. We have changed " pentagram " to " star". Moreover, we have added a comma before "and".

  1. Figure 6: Nodes with bootstrap support lower than 70 (I see just one) should be collapsed.

Answer: Thank you for your careful review. Nodes in ML tree with bootstrap support lower than 70 have been collapsed.

  1. l. 446. Please replace “as” with “in” and add “and” after “tree”.

Answer: Thank you for your careful review. We have replaced “as” with “in” and added “and” after “tree”.

Tables

  1. Table 1: l. 161. Maybe “asterisks” is missing here?

Answer: Thank you for your careful review. We have added a ✽ symbol in line 161.

Reviewer 2 Report

This article describes the mitogenomes of two shrimp species, Palaemon macrodactylus and P. tenuidactylus (Decapoda, Caridae), with the additional goals of investigating mitochondrial rearrangement within the family Palaeonomidae,  to clarify the taxonomic position, and resolve the phylogenetic relationships within the group. Additionally, the results provide valuable insight into the diversity of the mitochondrial genome. 

The following aspects should be revised and improved to make the article appealing to the broad audience of Genes:

1. Highlight the novel disciplinary contribution of the article that may contribute to the advance of taxonomy, phylogenetics, and systematics.  

2. Clarify why only two species were selected for sequencing the whole mitochondrial genome. The introduction mentions the lack of reports on the complete mitochondrial genomes of these two species but the conclusions highlight the need to incorporate similar studies in other related species.

3. If some of these species are of economic interest (Macrobrachium, for example), how this study could translate into practical benefits, if any?

Author Response

Response to Editors and Reviews

Manuscript ID: genes-2460626

Article Title: Comparison of mitochondrial genome sequences between two Palaemon species of the family Palaemonidae (Decapoda: Caridea): Gene Rearrangement and Phylogenetic implications

Dear Editors and Reviewers:

Thank you for your comments and suggestions on the paper. We have made efforts to clarify the following points and correct any mistakes. We sincerely appreciate the hard work of the editors and reviewers and hope that these revisions will be approved. On the following pages, please find the original comments from the reviewers (in black font) and our responses (in blue font). The revised manuscript has been completed, and all modifications have been made using the track changes mode. We hope that the revised manuscript meets the criteria for publication.

A point-by-point response to the comments:

Reviewer 2

This article describes the mitogenomes of two shrimp species, Palaemon macrodactylus and P. tenuidactylus (Decapoda, Caridae), with the additional goals of investigating mitochondrial rearrangement within the family Palaeonomidae,  to clarify the taxonomic position, and resolve the phylogenetic relationships within the group. Additionally, the results provide valuable insight into the diversity of the mitochondrial genome.

The following aspects should be revised and improved to make the article appealing to the broad audience of Genes:

Major comments:

  1. Highlight the novel disciplinary contribution of the article that may contribute to the advance of taxonomy, phylogenetics, and systematics.

Answer: Thank you for your suggestions. We greatly appreciate your comments as they have been instrumental in improving the quality of our article. We have supplemented the Introduction and Conclusions sections in the revised manuscript. The expanded availability of complete mitogenomes has the potential to aid in unraveling the phylogeny of Palaemonidae. This can be accomplished by offering multiple loci with varying rates of evolution, thus enhancing our understanding of their evolutionary relationships. Additionally, it allows for the expansion of taxonomic research methods, utilizing molecular techniques to provide more references for the classification of Palaemonidae.

  1. Clarify why only two species were selected for sequencing the whole mitochondrial genome. The introduction mentions the lack of reports on the complete mitochondrial genomes of these two species but the conclusions highlight the need to incorporate similar studies in other related species.

Answer: Thank you for your careful review. The Palaemonidae family is comprised of numerous economically valuable species and is one of the largest taxonomic units in the classification of true shrimps. However, the scope covered by the Palaemonidae family has been a subject of debate since its establishment. Mitochondrial genomes have been widely utilized in research areas such as population genetic structure, species identification, and systematic evolution. Therefore, the study of mitochondrial genomes in this family is highly necessary. Despite the ecological and economic importance of Palaemonidae species, the available mitogenome data for Palaemonidae is currently quite limited. In GenBank, there are only 18 complete mitogenomes available (as of July 5th, 2023, excluding unverified records) (https://www.ncbi.nlm.nih.gov/nuccore). However, due to the diverse habitats of the Palaemonidae family, sampling has been challenging, and at this stage, we were only able to obtain samples from two species within the Palaemon genus. Nonetheless, we will continue to focus on the classification of the Palaemon genus and strive to collect more species for further studies. The conclusion emphasizes the need for more mitochondrial genomes from the Palaemonidae family to be included in the research, which may have been stated in an imprecise manner and caused misunderstandings. We have removed this sentence from the revised manuscript. Instead, we emphasize the significance of sequencing the two species in this study, stating, "This study provides extensive information regarding the mitogenomes of Palaemon, laying a solid foundation for future research on genetic variation, systematic evolution, and breeding of Palaemon using mitogenomes."

  1. If some of these species are of economic interest (Macrobrachium, for example), how this study could translate into practical benefits, if any?

Answer: Thank you for your careful review. We have provided additional explanations for our work in the Introduction and Conclusion sections of the revised manuscript.As mentioned in the introduction, the Palaemonidae family has a wide range of habitats, and morphological identification and classification may not be sufficiently accurate. In tasks such as the introduction and breeding of these economically important species, known mitochondrial genome sequences can be of assistance. For example, the cox1 gene sequence, known as DNA barcoding, can help us utilize molecular techniques for species identification.

Reviewer 3 Report

The manuscript entitled “Comparison of mitochondrial genome sequences between two Palaemon species of the family Palaemonidae (Decapoda: Caridea): Gene Rearrangement and Phylogenetic implications” by Sun and colleagues describes two novel mitogenomes of Palaemon macrodactylus and Palaemon tenuidactylus and infers the phylogenetic relationships with other species belonging to infraorder Caridea.

Generally, the manuscript is quite well written and structured, even if I suggest some rearrangements in some paragraphs (see below). Moreover, I noticed several erroneous nomenclatures of the species that should be written always in italicus, as well as the genera. Please check all the document and correct them.

I suggest the following revisions:

-        Line 118 (and all manuscript): please use Italicus to refer to species name

-        Lines 137-138: I suggest to add also in this part the accession numbers of the two novel mitogenomes

-        Line 142: did Authors use the ORF Finder tool to determine boundaries among genes? If so, I suggest to add this information, otherwise I suggest to use it to be sure of them, e.g. for the overlapping between genes that they found or the truncated stop codon. Moreover, it is frequent in mitogenomes to have some intergenic base pairs between protein coding genes that could form a secondary structure in order to create a cleavage site of the polycistronic transcript.

-        Lines 199-204: I suggest to estimate the secondary structure of these intergenic spacers in order to detect eventually some cleavage site or putative regulatory regions. I would add this information and the secondary structure figures to the Paragraph 3.3

-        Lines 303-305: Could Authors discuss about the lack of the TΨC loop in the secondary structure of trnA, trnF, trnM and trnT? Did some other papers find out this lacking in other mitogenomes? Please include a discussion about this

-        Line 304: Please substitute trnF (reported twice) with trnT

Lines 325-335: I think that this part should be moved to Materials and Methods section

The manuscript is generally quite well written in English language

Author Response

Response to Editors and Reviews

Manuscript ID: genes-2460626

Article Title: Comparison of mitochondrial genome sequences between two Palaemon species of the family Palaemonidae (Decapoda: Caridea): Gene Rearrangement and Phylogenetic implications

Dear Editors and Reviewers:

Thank you for your comments and suggestions on the paper. We have made efforts to clarify the following points and correct any mistakes. We sincerely appreciate the hard work of the editors and reviewers and hope that these revisions will be approved. On the following pages, please find the original comments from the reviewers (in black font) and our responses (in blue font). The revised manuscript has been completed, and all modifications have been made using the track changes mode. We hope that the revised manuscript meets the criteria for publication.

A point-by-point response to the comments:

Reviewer 3

The manuscript entitled “Comparison of mitochondrial genome sequences between two Palaemon species of the family Palaemonidae (Decapoda: Caridea): Gene Rearrangement and Phylogenetic implications” by Sun and colleagues describes two novel mitogenomes of Palaemon macrodactylus and Palaemon tenuidactylus and infers the phylogenetic relationships with other species belonging to infraorder Caridea.

Generally, the manuscript is quite well written and structured, even if I suggest some rearrangements in some paragraphs (see below). Moreover, I noticed several erroneous nomenclatures of the species that should be written always in italicus, as well as the genera. Please check all the document and correct them.

I suggest the following revisions:

Major comments:

  1. Line 118 (and all manuscript): please use Italicus to refer to species name.

Answer: Thank you for your careful review. We have now changed all the species names to italics.

  1. Lines 137-138: I suggest to add also in this part the accession numbers of the two novel mitogenomes.

Answer: Thank you for your suggestion. We have added the accession numbers of two Palaemon species in this part.

  1. Line 142: did Authors use the ORF Finder tool to determine boundaries among genes? If so, I suggest to add this information, otherwise I suggest to use it to be sure of them, e.g. for the overlapping between genes that they found or the truncated stop codon. Moreover, it is frequent in mitogenomes to have some intergenic base pairs between protein coding genes that could form a secondary structure in order to create a cleavage site of the polycistronic transcript.

Answer: Thank you for your suggestion. We have used ORF Finder tool to determine boundaries among genes and added this information in the revised manuscript.

  1. Lines 199-204: I suggest to estimate the secondary structure of these intergenic spacers in order to detect eventually some cleavage site or putative regulatory regions. I would add this information and the secondary structure figures to the Paragraph 3.3.

Answer: Thank you for your suggestion. We attempted to predict the secondary structures of these intergenic regions, but we did not detect any eventually some cleavage site or putative regulatory regions.

  1. Lines 303-305: Could Authors discuss about the lack of the TΨC loop in the secondary structure of trnA, trnF, trnM and trnT? Did some other papers find out this lacking in other mitogenomes? Please include a discussion about this.

Answer: Thank you for your suggestion. We have discussed this aspect in the revised manuscript and inserted the corresponding reference.

  1. Line 304: Please substitute trnF (reported twice) with trnT.

Answer: Thank you for your careful review. We have already changed "trnF" to "trnT".

  1. Lines 325-335: I think that this part should be moved to Materials and Methods section.

Answer: Thank you for your suggestion. We have moved this section to the Materials and Methods.

Round 2

Reviewer 1 Report

Dear Authors,

thank you for your effort in improving and editing the original manuscript. While most concerns were correctly addressed, I still find that my major concern on phylogenetic reconstruction was not properly addressed.

At first, let me apologize if I was not able to explain myself: I did not mean to replace the phylogenetic reconstruction with the use of gene arrangement. I was suggesting to add such analysis to the main, large phylogenetic reconstruction. You are free to decide to include it or not in my opinion, this is not the central point.

The central point is the phylogenetic reconstruction itself: I do not feel it to be clearly described in the revised version of the manuscript. At l. 220 of the revised manuscript a single molecular evolution model is stated: TIM2+F+R10. Even if this is never clearly written, this leads me to understand that a single model was applied to a single, grand alignment. This is not a good approach in my opinion. Dataset should be partitioned according to single genes, and, possibly, to codon positions. Moreover, amino acid should be explored for the phylogenetic reconstruction. Instead, it is stated that sequences were "adjusted" (l. 209) using DAMBE: what does this mean? Then, it seems that all genes from a single species were concatenated (ll. 210-211) and that all these sequences (one for each species) were concatenated together (ll. 211-212). Maybe the point is the use of the verb "to concatenate". Maybe species-specific sequences were "included" in the same alignment? Then: how were sequences aligned? Gene by gene? Concatenation should take place after alignment and masking, and not before. Moreover, model selection should also take place at least for each gene, and not for the total alignment. This holds both for IQ-Tree and MrBayes. Finally, a partitioned, multimodel phylogenetic analysis has to be performed.

Best wishes!

Author Response

Response to Editors and Reviews

Manuscript ID: genes-2460626

Article Title: Comparison of mitochondrial genome sequences between two Palaemon species of the family Palaemonidae (Decapoda: Caridea): Gene Rearrangement and Phylogenetic implications

Dear Editors and Reviewers:

Thank you for your comments and suggestions on the paper. We have made efforts to clarify the following points and correct any mistakes. We sincerely appreciate the hard work of the editors and reviewers and hope that these revisions will be approved. On the following pages, please find the original comments from the reviewers (in black font) and our responses (in blue font). The revised manuscript has been completed, and all modifications have been made using the track changes mode. We hope that the revised manuscript meets the criteria for publication.

A point-by-point response to the comments:

Reviewer 1

The central point is the phylogenetic reconstruction itself: I do not feel it to be clearly described in the revised version of the manuscript. At l. 220 of the revised manuscript a single molecular evolution model is stated: TIM2+F+R10. Even if this is never clearly written, this leads me to understand that a single model was applied to a single, grand alignment. This is not a good approach in my opinion. Dataset should be partitioned according to single genes, and, possibly, to codon positions. Moreover, amino acid should be explored for the phylogenetic reconstruction. Instead, it is stated that sequences were "adjusted" (l. 209) using DAMBE: what does this mean? Then, it seems that all genes from a single species were concatenated (ll. 210-211) and that all these sequences (one for each species) were concatenated together (ll. 211-212). Maybe the point is the use of the verb "to concatenate". Maybe species-specific sequences were "included" in the same alignment? Then: how were sequences aligned? Gene by gene? Concatenation should take place after alignment and masking, and not before. Moreover, model selection should also take place at least for each gene, and not for the total alignment. This holds both for IQ-Tree and MrBayes. Finally, a partitioned, multimodel phylogenetic analysis has to be performed.

Answer: Thank you for your suggestion, and we apologize for the previous unclear description in the manuscript that led to your misunderstanding. We actually performed individual gene alignments followed by concatenation, and we also partitioned the data based on codon positions rather than using a single model. We have clarified and rewritten this section in the revised manuscript. Thank you again for your careful review. The revised content is as follows:

To explore the phylogenetic relationships within the Palaemonidae family, we downloaded sequences of 89 species from 12 Caridea families from GenBank (Table 1). We used the mitogenomes of Solenocera crassicornis (MF379621) and Metapenaeopsis dalei (NC_029457) from Dendrobranchiata as outgroups, and analyzed the phylogenetic re-lationships based on the 13 PCGs of these 93 species. We used DAMBE 7 software [43] to identify the sequence of the 13 PCGs from each sample. The nucleotide sequences for all 13 PCGs were individually aligned and masked using the default settings of ClustalW [44] in MEGA X, and then concatenated by PhyloSuite [51]. Afterward, Gblocks v.0.91b [52] was employed with default parameter settings to remove divergent and ambigu-ously aligned blocks, selecting conserved regions. The substitution saturation was cal-culated using the GTR substitution model via DAMBE 7, and the third position of the codons was excluded from subsequent analyses. We tested the selected DNA sequences for nucleotide models using jModelTest2.1.7 (https://code.google.com/p/jmodeltest2/) [53].

We employed two methods to analyze the phylogenetic relationships: the maxi-mum likelihood (ML) method using IQ-tree 2.1.3 [54], and the Bayesian inference (BI) method using MrBayes 3.2.7a [55]. Two partitions (first and second codon positions of 13PCGs) were set in the combined data set for partitioned Bayesian analyses using MrBayes v.3.2, we used PAUP 4 [56] for format conversion, and then used a combination of PAUP 4, ModelTest 3.7 [57], and MRModelTest 2.3 [58] software in MrMTgui to de-termine the best alternative model (GTR + I + G) according to the Akaike information criterion (AIC). The BI tree analysis was performed using four Markov Chain Monte Carlo (MCMC) chains simultaneously running for 2 million generations, with a sam-pling frequency of every 1000 generations. The first burn-in 25% of trees were discarded, and convergence for independent operation was evaluated using the mean standard deviation of the splitting frequency (<0.01). All parameters for effective sample size (ESS) were checked using Tracer v.1.7 [59]. For ML tree building with IQ-TREE, the same dataset was used. We used ModelFinder [60] to select the best alternative model (TIM2+F+R10) for the ML tree based on the Bayesian Information Criterion (BIC). The consensus tree was reconstructed, and 1000 ultrafast likelihood bootstrap replicates were utilized. Finally, we edited the phylogenetic tree using FigTree v1.4.3 [61].

Reviewer 2 Report

I appreciate the proper consideration of my queries and suggestions to improve the scientific quality and practical application of the article.

Author Response

Response to Editors and Reviews

Manuscript ID: genes-2460626

Article Title: Comparison of mitochondrial genome sequences between two Palaemon species of the family Palaemonidae (Decapoda: Caridea): Gene Rearrangement and Phylogenetic implications

Dear Editors and Reviewers:

Thank you for your comments and suggestions on the paper. We sincerely appreciate the hard work of the editors and reviewers.

Round 3

Reviewer 1 Report

I checked authors’ responses, but still there is something that should be fixed in my opinion. Below I report some concerns, using line numbers in the last, revised manuscript.

ll. 180-181. How is it possible to use DAMBE to “identify” sequences? Does this mean that authors used DAMBE to find ORFs? I guess that, at some stage, BLASTn or something similar was used to find similarities with annotated mitochondrial genes, and this should be clearly stated here.

l. 182. ClustalW is not a masking algorithm. How can sequences be “masked” using ClustalW? I guess that the masking step was performed using GBlocks (ll. 183-184).

ll. 188-189. Why was the third codon position excluded? Was it found to be saturated? This should be clearly described in the text.

ll. 196-201. This is the major issue, which leads me to think that a major revision is still required. Apparently, authors do not partitionate the dataset. For what I understand, they concatenated the complete dataset (13 PCGs), divided it into three partitions corresponding to the three codon positions, excluded the third codon position and then divided the remaining dataset into first codon position and second codon position (two partitions). A single molecular evolution model is provided at l. 200 (GTR+I+G), maybe because the same model was applied to both partitions. Instead, authors should divide the dataset into 13 partitions (the 13 PCGs) and then each of them should be divided into two partitions (first and second codon position), for a total of 26 partitions. Each of them should be tested for the best-performing molecular evolution model, and each of them should be passed to MB or IQ-Tree. Moreover, it is possible to infer phylogeny using aminoacids, and then the different phylogenies should be compared and contrasted to discuss the best phylogenetic reconstruction of Palaemonidae.

l. 209. Authors should insert a comma between “1” and “000”.

I thought that my previous reports were clear on these points, but maybe they were not, and I apologize for that. Do not hesitate to contact me if authors do not understand my concerns.

.

Author Response

Response to Editors and Reviews

Manuscript ID: genes-2460626

Article Title: Comparison of mitochondrial genome sequences between two Palaemon species of the family Palaemonidae (Decapoda: Caridea): Gene Rearrangement and Phylogenetic implications

Dear Editors and Reviewers:

Thank you for your comments and suggestions on the paper. We have made efforts to clarify the following points and correct any mistakes. We sincerely appreciate the hard work of the editors and reviewers and hope that these revisions will be approved. On the following pages, please find the original comments from the reviewers (in black font) and our responses (in blue font). The revised manuscript has been completed, and all modifications have been made using the track changes mode. We hope that the revised manuscript meets the criteria for publication.

A point-by-point response to the comments:

Reviewer 1

I checked authors’ responses, but still there is something that should be fixed in my opinion. Below I report some concerns, using line numbers in the last, revised manuscript.

  1. ll. 180-181. How is it possible to use DAMBE to “identify” sequences? Does this mean that authors used DAMBE to find ORFs? I guess that, at some stage, BLASTn or something similar was used to find similarities with annotated mitochondrial genes, and this should be clearly stated here.

Answer: Thank you for your review. It might have been a result of unclear expression, and we have now rephrased this step in the revised manuscript. This step refers to our process of downloading the complete mitochondrial genome sequences of 89 species from GenBank in gb format files and using DAMBE software to extract the sequences of each PCG, not to find ORFs. The procedure is illustrated in the figures below:

  1. 182. ClustalW is not a masking algorithm. How can sequences be “masked” using ClustalW? I guess that the masking step was performed using GBlocks (ll. 183-184).

Answer: Thank you for reviewing. We have made the revisions.

  1. ll. 188-189. Why was the third codon position excluded? Was it found to be saturated? This should be clearly described in the text.

Answer: Thank you for your careful review. The third codon position was excluded because it was found to be saturated. We have made revisions accordingly.

  1. ll. 196-201. This is the major issue, which leads me to think that a major revision is still required. Apparently, authors do not partitionate the dataset. For what I understand, they concatenated the complete dataset (13 PCGs), divided it into three partitions corresponding to the three codon positions, excluded the third codon position and then divided the remaining dataset into first codon position and second codon position (two partitions). A single molecular evolution model is provided at l. 200 (GTR+I+G), maybe because the same model was applied to both partitions. Instead, authors should divide the dataset into 13 partitions (the 13 PCGs) and then each of them should be divided into two partitions (first and second codon position), for a total of 26 partitions. Each of them should be tested for the best-performing molecular evolution model, and each of them should be passed to MB or IQ-Tree. Moreover, it is possible to infer phylogeny using aminoacids, and then the different phylogenies should be compared and contrasted to discuss the best phylogenetic reconstruction of Palaemonidae.

Answer: Thank you for your suggestion, which provides us with new research insights. However, we still have some questions. When you mentioned dividing the dataset into 13 partitions (13 PCGs), does it mean separating the same protein-coding genes of all species into individual partitions? Does each partition contain only the protein-coding genes of the same type? If that's the case, how can a single-gene model be applied to constructing a phylogenetic tree based on 13 PCGs? And how do we determine the best model when passing each partition to MB or IQ-Tree? Perhaps the reviewer was referring to tree-building methods using partial gene fragments or combinations of individual genes? Could you provide examples to clarify this?

In contrast, our approach is to construct a phylogenetic tree based on the 13 protein-coding genes of each species. The tree-building method we used aligns with many studies on phylogenetics based on complete mitochondrial genomes, where the 13 PCGs are concatenated and partitioned based on codon saturation [1-6]. We believe that constructing a tree based on the complete dataset of 13 PCGs should involve testing models using the entire PCGs dataset. It may be inaccurate to build a phylogenetic tree based on single-gene models for the 13 PCGs, but perhaps our understanding is flawed. We apologize if there are any misunderstandings. Once again, we sincerely appreciate your feedback and review!

References:

[1] Ji YT, Zhou XJ, Yang Q, Lu YB, Wang J, Zou JX. Adaptive evolution characteristics of mitochondrial genomes in genus Aparapotamon (Brachyura, Potamidae) of freshwater crabs. BMC Genomics. 2023 Apr 12;24(1):193. doi: 10.1186/s12864-023-09290-9. PMID: 37041498; PMCID: PMC10091551.

[2] Zhao B, Gao S, Zhao M, Lv H, Song J, Wang H, Zeng Q, Liu J. Mitochondrial genomic analyses provide new insights into the "missing" atp8 and adaptive evolution of Mytilidae. BMC Genomics. 2022 Nov 2;23(1):738. doi: 10.1186/s12864-022-08940-8. PMID: 36324074; PMCID: PMC9628169.

[3] Lu C, Huang X, Deng J. Mitochondrial genomes of soft scales (Hemiptera: Coccidae): features, structures and significance. BMC Genomics. 2023 Jan 21;24(1):37. doi: 10.1186/s12864-023-09131-9. PMID: 36670383; PMCID: PMC9863192.

[4] Li Y, Gu M, Liu X, Lin J, Jiang H, Song H, Xiao X, Zhou W. Sequencing and analysis of the complete mitochondrial genomes of Toona sinensis and Toona ciliata reveal evolutionary features of Toona. BMC Genomics. 2023 Feb 1;24(1):58. doi: 10.1186/s12864-023-09150-6. PMID: 36726084; PMCID: PMC9893635.

[5] Ji YT, Zhou XJ, Yang Q, Lu YB, Wang J, Zou JX. Adaptive evolution characteristics of mitochondrial genomes in genus Aparapotamon (Brachyura, Potamidae) of freshwater crabs. BMC Genomics. 2023 Apr 12;24(1):193. doi: 10.1186/s12864-023-09290-9. PMID: 37041498; PMCID: PMC10091551.

[6] Cheng Z L .Mitochondrial phylogenomics reveals insights into taxonomy and evolution of Penaeoidea (Crustacea: Decapoda)[J].Zoologica Scripta: An International Journal of Evolutionary Zoology, 2018, 47(5).

  1. 209. Authors should insert a comma between “1” and “000”.

Answer: We have inserted a comma between '1' and '000'.
